# Correlations and commuting transfer matrices in integrable unitary circuits

**Pieter W. Claeys$^\star$, Jonah Herzog-Arbeitman and Austen Lamacraft**

TCM Group, Cavendish Laboratory, University of Cambridge, Cambridge CB3 0HE, UK

$\star$ pc652@cam.ac.uk

## Abstract

We consider a unitary circuit where the underlying gates are chosen to be $\check{R}$-matrices satisfying the Yang-Baxter equation and correlation functions can be expressed through a transfer matrix formalism. These transfer matrices are no longer Hermitian and differ from the ones guaranteeing local conservation laws, but remain mutually commuting at different values of the spectral parameter defining the circuit. Exact eigenstates can still be constructed as a Bethe ansatz, but while these transfer matrices are diagonalizable in the inhomogeneous case, the homogeneous limit corresponds to an exceptional point where multiple eigenstates coalesce and Jordan blocks appear. Remarkably, the complete set of (generalized) eigenstates is only obtained when taking into account a combinatorial number of nontrivial vacuum states. In all cases, the Bethe equations reduce to those of the integrable spin-1 chain and exhibit a global $SU(2)$ symmetry, significantly reducing the total number of eigenstates required in the calculation of correlation functions. A similar construction is shown to hold for the calculation of out-of-time-order correlations.

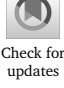

# 1  Introduction

In the past decade, the dynamics of out-of-equilibrium many-body quantum systems has become a major focus of both theoretical and experimental research [1–4]. In generic many-body systems local correlations are expected to thermalize after a sufficiently long time, where much understanding has been gained through the eigenstate thermalization hypothesis [4]. Still, it is in general extremely difficult to describe exact dynamics of many-body systems due to the rapid entanglement growth and resulting computational complexity [5–7].

It has recently been realized that so-called unitary circuits can provide a minimal model for many-body dynamics governed by local interactions, reproducing many features expected in generic models [6,8–13]. Unitary gates, originating in quantum computation, can generate 'dynamics' in discrete time through the repeated action on neighbouring sites in a lattice. Exact results include the characterization of entanglement growth [6,10,13], operator spreading [11], and reduced density matrices and the resulting entanglement [14]. The notion of unitary circuits has even been extended to take into account projective measurements, resulting in the appearance of a new kind of phase transition as the measurement rate is varied [15–18].

However, most exact results impose randomness on the underlying gates. If all gates are identical this results in Floquet dynamics, and away from the limit of a large local Hilbert space approximate methods need to be used [12]. In the past years various approaches have been developed to treat such circuits, either through an effective "entanglement membrane" [19], the local pairing of Feynman histories [20], or by an influence matrix based on the Keldysh path-integral formalism [21]. Another class of tractable models is those where the underlying Floquet circuits are composed of Clifford gates, which can be efficiently simulated classically [7], as used to study e.g. many-body localization [22]. While exact results remain relatively restricted, dual-unitary gates have recently emerged as a special class of circuits where correlations can be exactly calculated. As first identified in Ref. [23], within dual-unitary circuit models the only nontrivial correlations are those on the edge of the light cone, which can be exactly calculated using a quantum channel approach. Dual-unitary models have since been the subject of intensive study [14,24–35].

Another class for which exact results are available is that of integrable models, whose out-of-equilibrium dynamics has been studied in great detail (see e.g. [36] and Refs. therein). Integrable models are characterized by an extensive set of conserved charges, resulting in non-ergodic dynamics, and exact eigenstates can be obtained using Bethe ansatz techniques. Integrable systems exist as both local Hamiltonians and unitary circuits, where the latter can be interpreted as the Trotterization of the former [37–40]. In the case of unitary circuits, Bethe ansatz techniques have been used to calculate the spectral form factor for randomized Floquet models with a conserved charge [41] and (quasi)local conservation laws have been identified [37]. Diffusive transport has been observed in the (anisotropic) XXZ case [38], whereas superdiffusion and Kardar-Parisi-Zhang scaling is expected in (isotropic) XXX models [40,42–45].

We are often interested in studying quantities like correlation functions in the thermody-

namic limit of infinite system size. The usual approach to this limit is to evaluate the quantities of interest in a finite size system before taking the limit. The latter step is often highly involved, due to the exponentially growing number of contributions that have to be accounted for, so that some way of organizing these contributions must be found. In this paper we develop a transfer matrix formalism that exploits the unitary structure of the circuit to apply the techniques of integrability to infinite temperature correlation functions – directly in the thermodynamic limit. While the transfer matrix involved in these calculations has a more involved structure than the usual transfer matrix from integrability, its size is only determined by the distance of correlations from the causal light cone. This idea was introduced in Ref. [26], but we review it in the remainder of this introductory section in order to make the presentation self-contained before turning to the application of this technique to integrable systems.

Note that the integrability and associated (quasi)local conserved charges of these circuits has already been established by constructing a continuous family of commuting transfer matrices for each circuit [37]. These explicitly depend on the choice of spectral parameter in the $\check{R}$-matrix and hence the choice of circuit. The unitary evolution operators represented by two such different circuits do not commute, and have different conservation laws. Thus we would not expect the dynamics to be related. Remarkably, we find that the more involved transfer matrices in the calculation of correlations *do* commute for different choices of this spectral parameter, connecting the dynamics in seemingly unrelated circuits.

## 1.1 Correlation functions in unitary circuits

Unitary ciruits are constructed out of two-qubit gates. The underlying unitary gates $U$ and their hermitian conjugate $U^\dagger$ act on two copies of the local Hibert space, and can be graphically represented as

$$U_{ab,cd} = \vcenter{\hbox{\includegraphics{blue}}} \;, \qquad U^\dagger_{ab,cd} = \vcenter{\hbox{\includegraphics{red}}} \;. \tag{1}$$

In this graphical notation each leg carries a local 2-dimensional Hilbert space, and the indices of legs connecting two operators are implicitly summed over (see e.g. Ref. [46]). Unitarity is then graphically represented as

$$UU^\dagger = U^\dagger U = \mathbb{1} \qquad \Rightarrow \qquad \vcenter{\hbox{\includegraphics{gates}}} = \vcenter{\hbox{\includegraphics{gates2}}} = \vcenter{\hbox{\includegraphics{lines}}} \;. \tag{2}$$

These gates can be used to construct a unitary evolution operator in various ways. Here we focus on the 'brick circuit' dynamics: $\mathcal{U}(t)$ at time $t$ consists of the $t$-times repeated application of staggered two-site gates and can be graphically represented as

$$\mathcal{U}(t) = \vcenter{\hbox{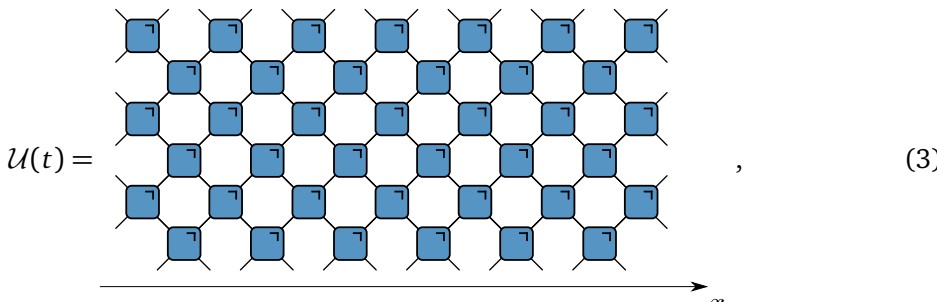}} \;, \tag{3}$$

where the total number of layers equals the discrete time $t$. We will consider correlation functions of one-site operators of the form

$$c_{\alpha\beta}(x,t) = \langle \sigma_\alpha(0,0)\sigma_\beta(x,t) \rangle = \langle \sigma_\alpha(0)\mathcal{U}(t)\sigma_\beta(x)\mathcal{U}^\dagger(t) \rangle, \tag{4}$$

at infinite temperature, i.e. $\langle \mathcal{O} \rangle = \mathrm{tr}(\mathcal{O})/\mathrm{tr}(\mathbb{1})$, and where the set $\{\sigma_\alpha, \alpha = 0, x, y, z\}$ represents the Pauli matrices with added $\sigma_0 = \mathbb{1}$. Due to the unitarity of the underlying gates, all correlation functions trivialize for $x > t$: unitary circuits have a built-in maximal correlation velocity [12]. For $x \leq t$, a nontrivial expression can be obtained for $c_{\alpha\beta}(x,t)$ as

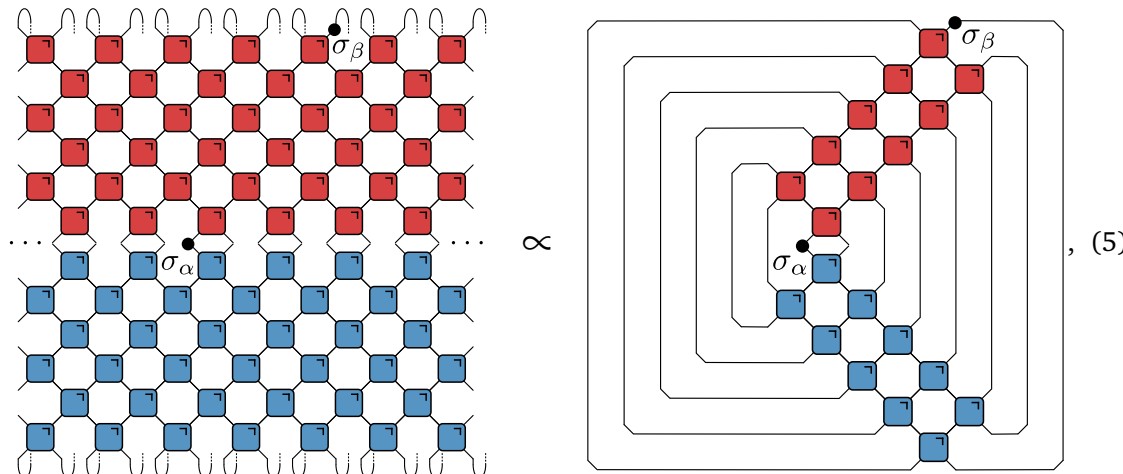

$$, \tag{5}$$

where the equality follows from the unitarity of the underlying gates.

No additional gates can be removed, but the final expression can be simplified to

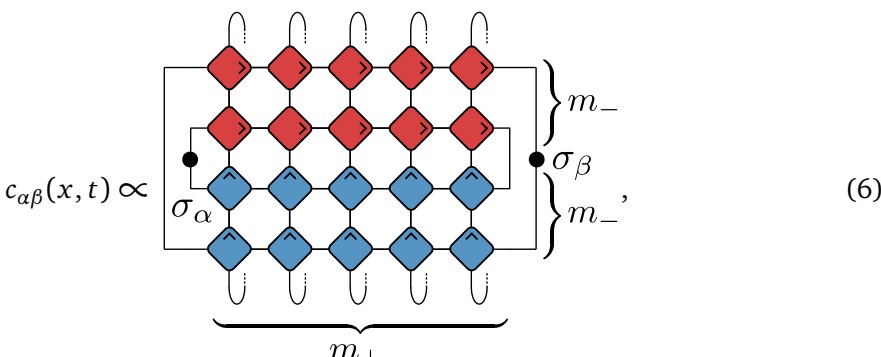

$$c_{\alpha\beta}(x,t) \propto \tag{6}$$

in which $m_+ = (t+x)/2$ and $m_- = (t-x+2)/2$. The prefactor does not depend on the specific operators $\sigma_{\alpha,\beta}$ and can always be recovered by demanding $c_{00}(x,t) = 1$. We can identify a transfer matrix

$$\tau_m = \frac{1}{2} \tag{7}$$

writing $m = m_-$ for ease of notation, and which we here rotated by 90 degrees for convenience. For a local 2-dimensional Hilbert space, this transfer matrix acts on a $4^m$-dimensional Hilbert space. Due to the exponential growth with $m = (t-x+2)/2$, explicit calculations are generally restricted to correlations close to the light cone or at short times. For $m = 1$ and hence $t = x$, this reduces to the quantum channel known to determine light-cone correlations [23, 26, 30]. Note that this transfer matrix does not act purely on space or on time, but rather evolves correlations in both dimensions along a fixed distance from the edge of the light cone. Within

the theory of integrable systems transfer matrices usually act on the full lattice [37,41], but here the size of the transfer matrix is independent of the system size and fully determined by $t - x$, the distance from the edge of the causal light cone. While we have implicitly assumed $t - x$ to be even in this derivation, the same transfer matrix appears for $t - x$ odd, only with slightly more involved boundary conditions (see e.g. Ref. [26]).

Being a transfer matrix, this operator is necessarily contracting and all eigenvalues $\lambda$ satisfy $|\lambda| \leq 1$. Furthermore, this operator has at least one eigenvalue $\lambda = 1$. Indeed, it can be checked that a trivial eigenstate/operator is given by

$$|\emptyset\rangle_m = \underbrace{\qquad\qquad}_{m}\underbrace{\qquad\qquad}_{m}, \tag{8}$$

which is here represented purely graphically. This eigenstate reflects the fact that unitary evolution acts trivially on the identity operator and $c_{00}(x, t) = 1, \forall x, t$ [1].

Reintroducing appropriate normalization constants, the correlations can be expressed as

$$c_{\alpha\beta}(n - m + 1, n + m - 1) = \left(\sigma_{\beta,m}\middle|\tau_m^n\middle|\sigma_{\alpha,m}\right), \tag{9}$$

with boundary conditions set by [2]

$$\left(\sigma_{\beta,m}\middle| = \frac{1}{2^{m/2}} \underbrace{\qquad\qquad}_{m}\underbrace{\qquad\qquad}_{m} \tag{10}$$

and

$$\middle|\sigma_{\alpha,m}\right) = \frac{1}{2^{m/2}} \underbrace{\qquad\qquad}_{m}\underbrace{\qquad\qquad}_{m}. \tag{11}$$

We emphasize that the discussion so far holds for arbitrary unitary gates. We now turn to the application of this formalism to gates that give rise to integrable circuits.

## 1.2 Outline

The outline of the remainder of this paper is as follows. In Section 2 we show, motivated by the Trotterization of the XXX Hamiltonian, how unitary gates may be chosen to be $\check{R}$-matrices satisfying the Yang-Baxter equation [47]. The resulting unitaries have a free (spectral) parameter, and we will see that the transfer matrices involved in computing the correlation functions, despite having the more involved structure apparent in Equation (7), still form a commuting family at different values of the spectral parameter, as in other integrable models. Next, the exact eigenstates of the transfer matrix at arbitrary distances from the light cone are found by applying the algebraic Bethe ansatz (ABA) construction. Proving the completeness of this construction turns out to be rather subtle, as the transfer matrix proves not to be diagonalizable. These subtleties also motivate our choice to be as explicit as possible in the derivation of all usual results from integrability. In Section 2.4 we introduce inhomogeneities into the model that preserve integrability and unitarity and allow us to demonstrate the completeness of the Bethe ansatz. In the homogeneous limit large Jordan blocks emerge, a situation which is analyzed in detail in Appendix A.1. Section 2.5 illustrates these results for small system sizes.

---

[1] Viewed as a quantum channel, the transfer matrix is *unital*.

[2] Note that, when acting from the top, the operator indices are labelled from the left to the right, while the reverse holds when acting from the bottom.

In Section 3 we extend our formalism to the calculation of out-of-time-order correlators (OTOCs) and explain the implications of integrability for these quantities. We conclude in Section 4 with an outlook for the prospects of extracting the asymptotic long-time behaviour of correlation functions using the formalism developed.

## 2 Correlation functions from integrability

### 2.1 Motivation: Trotterization of the XXX Hamiltonian

The unitary circuits considered in this paper can be motivated from the XXX Heisenberg Hamiltonian (see also Refs. [40, 41]) as given by

$$H = \sum_n h_{n,n+1} = -J \sum_n \left( \sigma_x^n \sigma_x^{n+1} + \sigma_y^n \sigma_y^{n+1} + \sigma_z^n \sigma_z^{n+1} \right). \tag{12}$$

The resulting unitary evolution operator $\exp[-iHt]$ can be Trotter decomposed as

$$\exp[-iHt] = \lim_{\delta t \to 0} \left[ \exp\left( -i\delta t \sum_{n\,\text{even}} h_{n,n+1} \right) \exp\left( -i\delta t \sum_{n\,\text{odd}} h_{n,n+1} \right) \right]^{(t/\delta t)}. \tag{13}$$

The resulting unitary evolution can be exactly represented as a brick circuit of the form (3), since

$$\exp\left( -i\delta t \sum_{n\,\text{even}} h_{n,n+1} \right) = \prod_{n\,\text{even}} e^{iJ\delta t} \check{R}_{n,n+1}(\tan 2J\delta t), \tag{14}$$

$$\exp\left( -i\delta t \sum_{n\,\text{odd}} h_{n,n+1} \right) = \prod_{n\,\text{odd}} e^{iJ\delta t} \check{R}_{n,n+1}(\tan 2J\delta t), \tag{15}$$

where the unitary gates acting on neighbouring sites are

$$\check{R}(\lambda) = \;\;\boxed{\lambda}\;\; = \frac{\mathbb{1} + i\lambda P}{1 + i\lambda}, \tag{16}$$

with $P$ the permutation operator. The matrix elements are explicitly given by

$$\check{R}(\lambda)_{ab,cd} = \;\;\overset{a \quad b}{\underset{c \quad d}{\boxed{\lambda}}}\;\; = \frac{1}{1 + i\lambda} \left( \delta_{ac}\delta_{bd} + i\lambda\delta_{ad}\delta_{bc} \right). \tag{17}$$

Note that $\check{R}^{-1}(\lambda) = \check{R}(-\lambda)$. Unitary matrices correspond to $\lambda$ real, in which case $\check{R}^\dagger(\lambda) = \check{R}(-\lambda)$, which we denote graphically by

$$\check{R}^\dagger(\lambda) = \check{R}(-\lambda) \quad \Rightarrow \quad \boxed{\lambda} = \boxed{-\lambda}. \tag{18}$$

By keeping the time step $\delta t$ finite, we can ask about the dynamics of the circuit in its own right. Remarkably, the dynamics is integrable at finite $\delta t$ [40, 41], as we now show.

## 2.2 Commuting transfer matrices

The key property that underlies the integrability of the above circuit is the famous braiding relation

$$\check{R}_{12}(\lambda)\check{R}_{23}(\lambda+\mu)\check{R}_{12}(\mu) = \check{R}_{23}(\mu)\check{R}_{12}(\lambda+\mu)\check{R}_{23}(\lambda),\tag{19}$$

where the overall expression acts on three copies of the local Hilbert space and $\check{R}_{ij}$ acts on the $i$-th and $j$-th copy. This braiding relation is equivalent to the Yang-Baxter equation, since the $\check{R}$-matrix is related through the $R$-matrix of integrability through $\check{R} = PR$, with the $R$-matrix satisfying the Yang-Baxter equation.

Graphically, the braiding relation can be expressed as

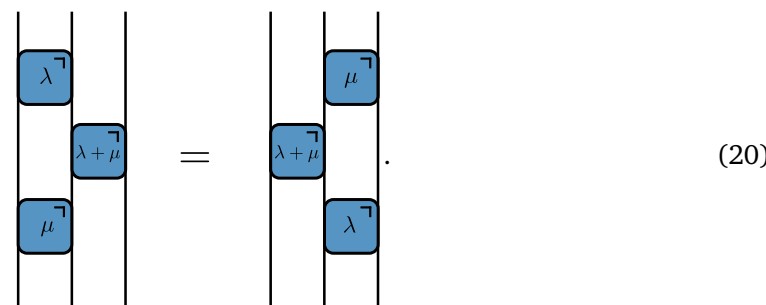

$$\tag{20}$$

Making use of Eq. (18), the transfer matrix (7) can be rewritten as

$$\tau_m(\lambda) = \underbrace{\qquad}_{m}\underbrace{\qquad}_{m},\tag{21}$$

where we have made the dependence on the parameter $\lambda$ explicit. From the graphical notation the matrix elements of $\tau_m(\lambda) \in \mathbb{C}^{2m\times 2m}$ follow as

$$\tau_m(\lambda)_{i_1\dots i_{2m},j_1\dots j_{2m}} = \qquad,\tag{22}$$

where the implicit summations can be made explicit as

$$\tau_m(\lambda)_{i_1\dots i_{2m},j_1\dots j_{2m}} = \sum_{a_1\dots a_{2m}}\qquad.\tag{23}$$

Written out purely in terms of the underlying $\check{R}$-matrices, the matrix elements of the transfer matrix follow as

$$\tau_m(\lambda)_{i_1\dots i_m,j_1\dots j_m} = \sum_{a_1\dots a_{2m}} \check{R}(-\lambda)_{a_1 i_1,j_1 a_2}\check{R}(-\lambda)_{a_2 i_2,j_2 a_3}\dots\check{R}(-\lambda)_{a_m i_m,j_m a_{m+1}}$$

$$\times \check{R}(\lambda)_{j_{m+1}a_{m+1},a_{m+2}i_{m+1}}\check{R}(\lambda)_{j_{m+2}a_{m+2},a_{m+3}i_{m+2}}\dots\check{R}(\lambda)_{j_{2m}a_{2m},a_1 i_{2m}}.\tag{24}$$

The resulting transfer matrices differ in some important ways from the ones usually encountered in the literature on integrability: they now exhibit a 'two-component' structure

where the orientation of the underlying gates changes halfway through, from which it can be easily checked that they are not Hermitian and as such not guaranteed to be diagonalizable. While the situation where a generic non-Hermitian matrix is nondiagonalizable is rare, the homogeneous model where all unitary gates in the transfer matrix are identical will be shown to be an exceptional point. No eigendecomposition exists, and a Jordan decomposition with nontrivial Jordan blocks should rather be considered.

Despite these differences, the transfer matrix retains one of the crucial properties of integrability. Namely, transfer matrices evaluated at different parameters commute

$$[\tau_m(\lambda), \tau_m(\mu)] = 0, \qquad \forall \lambda, \mu, \tag{25}$$

giving rise to a family of conserved charges. For hermitian operators this would imply that they can be simultaneously diagonalized, but here this only implies that they can be simultaneously Schur decomposed in upper-triangular matrices. While the commutativity of these transfer matrices might not seem surprising, note that the conserved charges for the unitary circuits (3) depend explicitly on the choice of $\lambda$. The two transfer matrices $\tau_m(\lambda)$ and $\tau_m(\mu)$ determine the correlations for circuits with different conservation laws, i.e. the transfer matrices generating the conserved charges for the circuits built out of $R(\lambda)$ and $R(\mu)$ do not commute if $|\lambda| \neq |\mu|$ [37].

Let us see how commuting transfer matrices are a direct consequence of the braiding relation. We can write the transfer matrix as the trace of a so-called monodromy matrix $T_m(\lambda)$ as

$$\tau_m(\lambda) = \mathrm{Tr}[T_m(\lambda)], \tag{26}$$

where the trace is being taken over the horizontal indices and

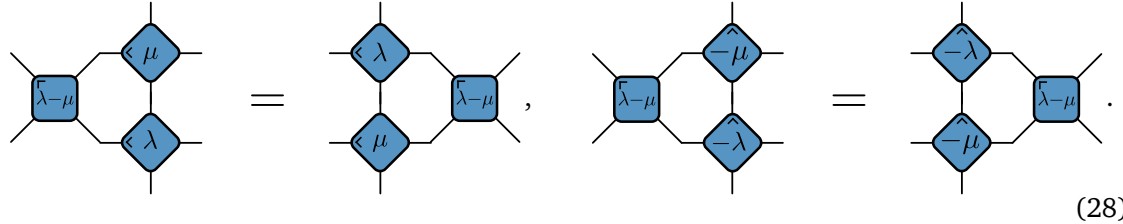

$$\tag{27}$$

In keeping with the usual terminology of the algebraic Bethe ansatz, we will call the space corresponding to the horizontal indices the 'auxiliary space', though it is part of the physical Hilbert space. Note that we could have written $T_m(\lambda)$ in different ways by breaking the horizontal legs at different points.

The braiding relation (20) can be recast in two different ways as

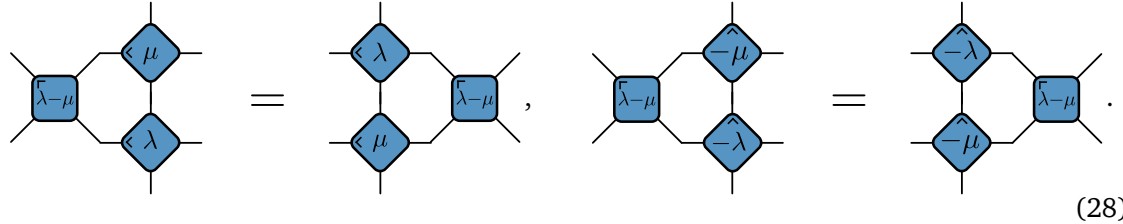

$$\tag{28}$$

In parsing these graphical expressions it is useful to bear in mind that rotating a gate through $180°$ leaves it unchanged.

These two relations immediately result in the so-called RTT relation for the monodromy matrix

$$\check{R}_{12}(\lambda - \mu) T_m(\lambda)_1 T_m(\mu)_2 = T_m(\mu)_2 T_m(\lambda)_1 \check{R}_{12}(\lambda - \mu)$$
$$\Rightarrow \quad T_m(\lambda)_1 T_m(\mu)_2 = \check{R}_{12}(\mu - \lambda) T_m(\mu)_2 T_m(\lambda)_1 \check{R}_{12}(\lambda - \mu), \tag{29}$$

where we have added a subscript to all operators in order to make clear which of the two copies of the auxiliary space they act on. Graphically, the RTT relation is represented as

$$
\begin{array}{cccc}
\check{R}_{\lambda-\mu} \, T_m(\mu) \, T_m(\lambda) & = & T_m(\lambda) \, T_m(\mu) \, \check{R}_{\lambda-\mu} & .
\end{array}
\tag{30}
$$

After multiplying on the left or right with $\check{R}^{\dagger}(\lambda - \mu)$ and taking the trace over the auxiliary spaces, the cyclic property of the trace returns $\tau_m(\lambda)\tau_m(\mu) = \tau_m(\mu)\tau_m(\lambda)$.

This commutativity can also be understood by noting that the change in orientation of the $\check{R}$-matrices can be absorbed in a local unitary transformation, since

$$
\sigma_y \, \check{R}_{-\lambda} \, \sigma_y = \frac{-i\lambda}{1 - i\lambda} \, \check{R}_{\lambda+i} \, .
\tag{31}
$$

As such, a local unitary transformation by $\sigma^y$ on the left $m$ sites in the transfer matrix returns the usual transfer matrix from integrability, with all gates having the same orientation, at the cost of introducing inhomogeneities, setting $\lambda \to \lambda + i$ in half the sites. While inhomogeneities often do not influence the properties of the transfer matrices, inhomogeneities $i$ are known to be a pathological case, e.g. preventing the usual proofs of completeness for the Bethe ansatz [48]. This can be understood since the braiding relation is the fundamental relation underlying all properties of integrable systems, which becomes singular when braiding two gates parameterized by $\mu \to \lambda + i$ and $\lambda$; $\lim_{\mu \to \lambda+i} \check{R}(\mu - \lambda)$ is singular. In the following we will stick with the original notation, since the additional rotation also obscures the fact that the unitary circuits are related by hermitian conjugation, which will be important in the following.

## 2.3 Bethe states

While the previous section presented a straightforward extension of the usual construction for commuting transfer matrices, the construction of eigenstates is more subtle. Within the algebraic Bethe ansatz (ABA), exact Bethe eigenstates require the existence of (i) generalized commutation relations between the 'matrix elements' of the monodromy matrix and (ii) a vacuum state.

The first directly follows from the RTT relations in the usual way, where the generalized commutation relations are only determined by the initial choice of $R$-matrix. Writing out the matrix elements of $T_m(\lambda)$ in the auxiliary space

$$
T_m(\lambda) = \begin{bmatrix} T_{00}(\lambda) & T_{01}(\lambda) \\ T_{10}(\lambda) & T_{11}(\lambda) \end{bmatrix} = \begin{bmatrix} A(\lambda) & B(\lambda) \\ C(\lambda) & D(\lambda) \end{bmatrix},
\tag{32}
$$

the $RTT$ relation implies generalized commutations relations. These can be found in e.g. Refs. [47, 49]. In the following, we will need

$$
A(\lambda)B(\mu) = f(\mu, \lambda)B(\mu)A(\lambda) + g(\lambda, \mu)B(\lambda)A(\mu),
\tag{33}
$$

$$
D(\lambda)B(\mu) = f(\lambda, \mu)B(\mu)D(\lambda) + g(\mu, \lambda)B(\lambda)D(\mu),
\tag{34}
$$

$$[B(\lambda), B(\mu)] = 0, \tag{35}$$

with

$$f(\lambda, \mu) = 1 + \frac{i}{\lambda - \mu}, \qquad g(\lambda, \mu) = \frac{i}{\lambda - \mu}. \tag{36}$$

For the second part, the (pseudo-)vacuum state $|\emptyset\rangle$ needs to be an eigenstate of $A(\lambda)$ and $D(\lambda)$ and annihilated by $C(\lambda)$, satisfying

$$A(\lambda)|\emptyset\rangle = a(\lambda)|\emptyset\rangle, \qquad D(\lambda)|\emptyset\rangle = d(\lambda)|\emptyset\rangle, \qquad C(\lambda)|\emptyset\rangle = 0. \tag{37}$$

One such vacuum state can immediately be found as $|\emptyset\rangle_m = |\underbrace{0\ldots 0}_{m}\underbrace{1\ldots 1}_{m}\rangle$, where

$$a(\lambda) = \left(\frac{i\lambda}{1 + i\lambda}\right)^m, \qquad d(\lambda) = \left(\frac{-i\lambda}{1 - i\lambda}\right)^m. \tag{38}$$

Making use of the usual Bethe ansatz construction, this returns eigenstates

$$|\{\lambda_1, \ldots, \lambda_N\}\rangle = \prod_{k=1}^{N} B(\lambda_k)|\emptyset\rangle_m, \tag{39}$$

with $\tau_m(\mu)$ having eigenvalues dependent on the *spectral parameter* $\mu$

$$t_m(\mu|\lambda_1\ldots\lambda_N) = a(\mu)\prod_{k=1}^{N} f(\lambda_k, \mu) + d(\mu)\prod_{k=1}^{N} f(\mu, \lambda_k) \tag{40}$$

$$= \left(\frac{i\mu}{1 + i\mu}\right)^m \prod_{k=1}^{N}\left(1 + \frac{i}{\lambda_k - \mu}\right) + \left(\frac{-i\mu}{1 - i\mu}\right)^m \prod_{k=1}^{N}\left(1 - \frac{i}{\lambda_k - \mu}\right), \tag{41}$$

provided the Bethe equations are satisfied,

$$\left(\frac{\lambda_j - i}{\lambda_j + i}\right)^m = \prod_{k \neq j}^{N} \frac{\lambda_k - \lambda_j + i}{\lambda_k - \lambda_j - i}. \tag{42}$$

These can be compared with the usual Bethe equation for integrable spin-$s$ chains [50,51],

$$\left(\frac{\lambda_j - is}{\lambda_j + is}\right)^m = \prod_{k \neq j}^{N} \frac{\lambda_k - \lambda_j + i}{\lambda_k - \lambda_j - i}. \tag{43}$$

Remarkably, the obtained equations are exactly the Bethe equations for the integrable spin-1 Babujan-Takhtajan chain with $m$ sites [52–54]. The reason for this correspondence will be detailed in Section 2.4. These Bethe equation have been extensively studied in the literature, and satisfy the string hypothesis in the thermodynamic limit (see e.g. [55,56]). From studies of the spin-1 chain they are also known to be complete, such that there exist $3^m$ distinct solutions to the Bethe equations and hence $3^m$ eigenstates of the transfer matrix (7) can be obtained in this way[3]. This immediately implies the incompleteness of the Bethe ansatz for the current transfer matrix, which acts on a $4^m$ dimensional Hilbert space: only an exponentially small fraction of the full Hilbert space can be written as Eq. (39).

Additional eigenstates can be obtained by noting that $|\emptyset\rangle_m$ is not the only state satisfying the requirements of a pseudo-vacuum state. In fact, an extensive number of pseudo-vacuum

---

[3]Including possible diverging rapidities resulting in raising operators for the global $SU(2)$ symmetry, see Appendix A.2.

states can be obtained by making use of the two-component structure of the transfer matrix, where the central unitary gates are related through hermitian conjugation. Making use of the unitarity, we observe that

$$
\begin{array}{c}
\text{(44)}
\end{array}
$$

returning $T_{m-1}$ from $T_m$, such that any vacuum state for $T_{m-1}$ can be used to construct a vacuum state for $T_m$. This relation can be repeatedly applied, and a total number of $m$ orthogonal pseudo-vacuum states can be defined as

$$
|\emptyset\rangle_{m,n} = \Big| \underbrace{0 \quad\quad 0}_{n} \quad \underbrace{\underline{\quad\lfloor\ \ \rfloor\quad}}_{m-n} \quad \underbrace{1 \quad\quad 1}_{n} \Big\rangle , \tag{45}
$$

with $n = 0,\dots,m$. Here $|\emptyset\rangle_{m,m}$ returns the previously defined vacuum state and $|\emptyset\rangle_{m,0}$ is a trivial eigenstate following from the total unitarity of the circuit. These states are again annihilated by $C(\lambda)$ and satisfy

$$
A(\lambda)|\emptyset\rangle_{m,n} = \left(\frac{i\lambda}{1+i\lambda}\right)^n |\emptyset\rangle_{m,n}, \qquad D(\lambda)|\emptyset\rangle_{m,n} = \left(\frac{-i\lambda}{1-i\lambda}\right)^n |\emptyset\rangle_{m,n}. \tag{46}
$$

Each state $|\emptyset\rangle_{m,n}$ returns an additional $3^n$ eigenstates of $\tau_m(\lambda)$,

$$
|\{\lambda_1,\dots,\lambda_N\}\rangle = \prod_{k=1}^{N} B(\lambda_k)|\emptyset\rangle_{m,n}, \tag{47}
$$

with eigenvalues

$$
t_m(\mu|\lambda_1\dots\lambda_N) = \left(\frac{i\mu}{1+i\mu}\right)^n \prod_{k=1}^{N}\left(1+\frac{i}{\lambda_k-\mu}\right) + \left(\frac{-i\mu}{1-i\mu}\right)^n \prod_{k=1}^{N}\left(1-\frac{i}{\lambda_k-\mu}\right), \tag{48}
$$

provided the Bethe equations are satisfied,

$$
\left(\frac{\lambda_j-i}{\lambda_j+i}\right)^n = \prod_{k\neq j}^{N} \frac{\lambda_k-\lambda_j+i}{\lambda_k-\lambda_j-i}. \tag{49}
$$

This results in a nested structure, where each Bethe state for the spin-1 chain with $n \leq m$ sites returns an eigenstate of the transfer matrix for the unitary circuit containing $2m$ sites, and the spectrum of $\tau_n(\lambda)$ is contained in the spectrum of $\tau_m(\lambda), n \leq m$. Note that the latter does not depend on the integrability of the circuit: these contractions only depend on the unitarity of the individual gates and such nesting will occur for general unitary circuits.

Alternatively, these states can be recast as Bethe states constructed using the initial vacuum state $|\emptyset\rangle_m = |\emptyset\rangle_{m,m}$. The repeated action of $B(0)$ on a vacuum state returns additional vacuum states,

$$
B(0)|\emptyset\rangle_m = |\emptyset\rangle_{m,m-1}, \qquad B(0)|\emptyset\rangle_{m,m-1} = |\emptyset\rangle_{m,m-2}, \qquad \dots \tag{50}
$$

Mathematically, these states don't naturally arise as a solution to the Bethe equations because we divided by $\lambda_j$ in order to recast these equations in their usual form, such that these additional solutions need to be explicitly taken into account.

This now returns a total of

$$\sum_{n=0}^{m} 3^n = 3^{m+1} - 1 \tag{51}$$

distinct eigenstates (which might be reduced by Bethe states where a subset of the rapidities equal zero, leading to coinciding eigenstates). While the additional vacuum states increase the total number of Bethe eigenstates, these still only span an exponentially small fraction of the total Hilbert space with dimension $4^m$. However, this already fully exploits the unitarity of the underlying gates, and no other pseudo-vacuum states can be constructed in this way.

This could imply that there are either additional eigenvectors that cannot be written as a Bethe ansatz, or we have exhausted all eigenstates and need to consider Jordan blocks of generalized eigenvectors. In the following, we will show that the second scenario holds.

## 2.4 The inhomogeneous model

The issue of completeness can be circumvented by introducing inhomogeneities in the initial circuit. Taking all circuits to be equal significantly simplifies the resulting expressions, but conceals an additional structure in the inhomogeneous circuit. Readers only interested in the homogeneous model can skip forward to Section 2.5 for a discussion of the eigenvalues and dimensions of the Jordan blocks in specific transfer matrices. In this section, we will associate an inhomogeneity $\epsilon_i \in \mathbb{R}$ with each unitary gate, now given by $\check{R}(\lambda - \epsilon_i)$, such that $m$ inhomogeneities $\{\epsilon_1, \ldots, \epsilon_m\}$ appear in $\tau_m(\lambda)$.

With the inclusion of inhomogeneities in the circuit, a similar calculation as for (7) returns a monodromy matrix

$$T_m(\lambda | \epsilon_1 \ldots \epsilon_m) = \tag{52}$$

The inhomogeneities on both sides of this monodromy matrix are necessarily related through the underlying unitary circuit construction, which will be important in the following.

The RTT relation still holds, such that transfer matrices with different spectral parameters commute as long as the inhomogeneities are identical,

$$[\tau_m(\lambda | \epsilon_1 \ldots \epsilon_m), \tau_m(\mu | \epsilon_1 \ldots \epsilon_m)] = 0, \qquad \forall \lambda, \mu. \tag{53}$$

Similarly, the Bethe ansatz construction still holds since the inhomogeneities only influence the eigenvalues of the pseudo-vacuum states. The states $|\emptyset\rangle_{m,n}$ still satisfy the requirements for vacuum states, and the Bethe equations are explicitly given by

$$\prod_{l=1}^{n} \frac{\lambda_j - \epsilon_l - i}{\lambda_j - \epsilon_l + i} = \prod_{k \neq j}^{N} \frac{\lambda_k - \lambda_j + i}{\lambda_k - \lambda_j - i}, \tag{54}$$

leading to eigenvalues

$$\prod_{l=1}^{n} \left( \frac{i(\mu - \epsilon_l)}{1 + i(\mu - \epsilon_l)} \right) \prod_{k=1}^{N} \left( 1 + \frac{i}{\lambda_k - \mu} \right) + \prod_{l=1}^{n} \left( \frac{-i(\mu - \epsilon_l)}{1 - i(\mu - \epsilon_l)} \right) \prod_{k=1}^{N} \left( 1 - \frac{i}{\lambda_k - \mu} \right). \tag{55}$$

Note that the transfer matrix and its eigenstates depend not just on the choice of inhomogeneities, but also on their ordering: $\tau_m(\lambda | \epsilon_1 \epsilon_2 \ldots) \neq \tau_m(\lambda | \epsilon_2 \epsilon_1 \ldots)$. However, as is common in the ABA, the spectrum does not depend on the choice of ordering. Any permutation of

the inhomogeneities corresponds to a unitary transformation of the transfer matrix, leaving the spectrum invariant, where inhomogeneities can be pairwise exchanged through a unitary transformation with an $\check{R}$-matrix – again making use of the braiding relation. This is illustrated below for $\epsilon_1$ and $\epsilon_2$ in a transfer matrix with $m = 3$,

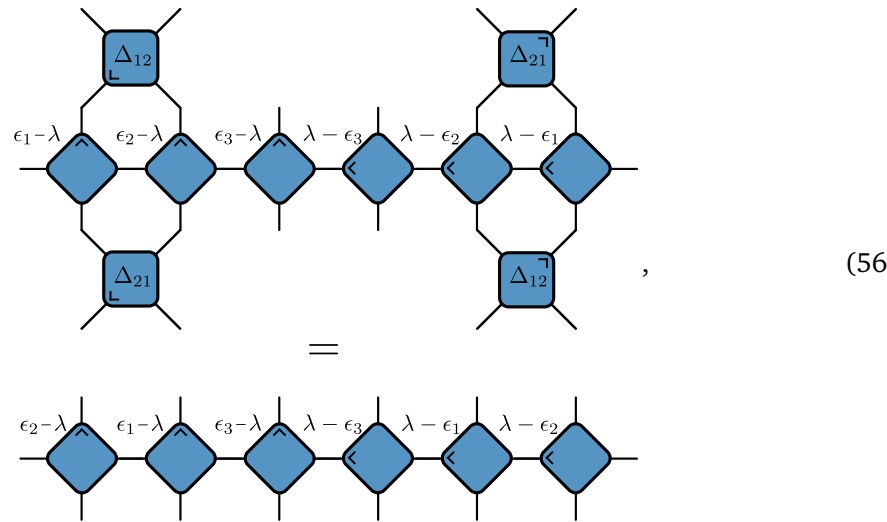

$$\text{(56)}$$

with $\Delta_{12} = \epsilon_1 - \epsilon_2 = -\Delta_{21}$. In this way, any permutation $P$ on $m$ elements can be associated with a unitary operator such that

$$T_m(\lambda|\epsilon_1\ldots\epsilon_m) = U_P^\dagger T_m(\lambda|\epsilon_{P(1)}\ldots\epsilon_{P(m)})U_P\,, \tag{57}$$

implying

$$\tau_m(\lambda|\epsilon_1\ldots\epsilon_m) = U_P^\dagger \tau_m(\lambda|\epsilon_{P(1)}\ldots\epsilon_{P(m)})U_P\,. \tag{58}$$

In the usual Bethe ansatz construction, these unitary transformations act trivially on the eigenstates, exchanging the inhomogeneities in the generalized raising operators and leaving the vacuum state invariant. However, this is not the case for the contracted vacuum states $|\emptyset\rangle_{m,0<n<m}$. This can easily be seen from the action of $\tau(\lambda|\epsilon_1\ldots\epsilon_m)$ on $|\emptyset\rangle_{m,n}$. The $m-n$ contractions in this state remove any dependence on the inhomogeneities $\epsilon_{n+1},\ldots,\epsilon_m$, such that the eigenvalue will only depend on $\epsilon_1,\ldots,\epsilon_n$. We can write

$$\tau_m(\lambda|\epsilon_1\ldots\epsilon_m)|\emptyset\rangle_{m,n} = t_m(\lambda|\epsilon_1\ldots\epsilon_n)|\emptyset\rangle_{m,n}\,, \tag{59}$$

where we have introduced a shorthand notation $t_m(\lambda|\epsilon_1\ldots\epsilon_n)$ for the eigenvalue associated with the pseudo-vacuum $|\emptyset\rangle_{m,n}$. For $n < m$ this makes clear that all dependence on $\epsilon_{n+1},\ldots,\epsilon_m$ is lost.

Reordering the inhomogeneities through a unitary transformation will result in a dependence on $\epsilon_{P(1)},\ldots,\epsilon_{P(n)}$, such that acting with $U_P$ on $|\emptyset\rangle_{m,n<m}$ necessarily returns a different eigenstate with a different eigenvalue. Such a state remains an eigenstate, since

$$\begin{aligned} \tau_m(\lambda|\epsilon_1\ldots\epsilon_m)U_P|\emptyset\rangle_{m,n} &= U_P\tau_m(\lambda|\epsilon_{P(1)}\ldots\epsilon_{P(m)})|\emptyset\rangle_{m,n} \\ &= t_m(\lambda|\epsilon_{P(1)}\ldots\epsilon_{P(n)})U_P|\emptyset\rangle_{m,n}\,. \end{aligned} \tag{60}$$

Similar relations hold for all components of the transfer matrix, such that we can conclude that additional pseudo-vacuum states can be generated by acting with unitary transformation $U_P$ on the pseudo-vacuum states for the homogeneous model $|\emptyset\rangle_{m,n}$. Within the usual Bethe construction, such unitary transformations reordering the inhomogeneities act trivially on the vacuum state, but the two-component structure in the current model provides pseudo-vacuum states on which such unitary transformations act nontrivially.

We can associate a different unitary transformation with each permutation of $m$ elements. However, not every unitary operator returns a different state when acting on a pseudo-vacuum $|\emptyset\rangle_{m,n}$, as illustrated by the following two examples:

1. For $|\emptyset\rangle_{m,0}$ (contracting all indices in both components), we can make use of the unitarity of $\check{R}$-matrices on both sides, since

$$\text{(61)}$$

such that $U_P |\emptyset\rangle_{m,0} = |\emptyset\rangle_{m,0}$ for all permutations $P$.

2. For $|\emptyset\rangle_{m,m}$, containing no contractions, we use the fact that the $\check{R}$-matrix conserves spin [c.f. Equation (16)]: $\check{R}(\lambda)_{ab,00} = \delta_{ab,00}$ and $\check{R}(\lambda)_{ab,11} = \delta_{ab,11}$, to show that

$$\text{(62)}$$

such that $U_P |\emptyset\rangle_{m,m} = |\emptyset\rangle_{m,m}$ for all permutations $P$.

More generally, the state $U_P |\emptyset\rangle_{m,m} = |\emptyset\rangle_{m,m}$ is invariant under permutations among the $m-n$ contracted sites, and under permutations among the $n$ uncontracted sites, leading to $\binom{m}{n}$ distinct pseudo-vacuum states. This invariance is a direct result of the above identities combined with the braiding relation (20). While straightforward to show, the graphical proof is quite involved. We will explicitly show this equivalence for a small transfer matrix containing 3 inhomogeneities, which can then be immediately extended to transfer matrices of arbitrary size.

**Exchanging uncontracted sites.** Consider a transfer matrix depending on inhomogeneities $\{\epsilon_1, \epsilon_2, \epsilon_3\}$ and the vacuum state with $n = 2$. For any permutation, which we note as $(P(1), P(2), P(3))$, the single contraction in the vacuum state removes any dependence on $\epsilon_{P(3)}$ in the eigenstates and eigenvalue. Additional permutations exchanging the first two inhomogeneities leave the vacuum state $U_P |\emptyset\rangle_{3,2}$ invariant. Assume we want to remove the dependence on $\epsilon_1$, leading to an additional vacuum state with eigenvalues and eigenstates only dependent on $\epsilon_2$ and $\epsilon_3$. Any permutation can be constructed through an elementwise exchanges of neighbouring elements, and we have two possible permutations $(2, 3, 1)$ and $(3, 2, 1)$ satisfying $\epsilon_{P(3)} = \epsilon_1$. These permutations can be constructed as

$$(1, 2, 3) \overset{(12)}{\to} (2, 1, 3) \overset{(13)}{\to} (2, 3, 1), \tag{63}$$

$$(1, 2, 3) \overset{(23)}{\to} (1, 3, 2) \overset{(13)}{\to} (3, 1, 2) \overset{(12)}{\to} (3, 2, 1), \tag{64}$$

with two different permutations only differing in the first two elements, corresponding to the uncontracted inhomogeneities. If each permutation would result in a unique pseudo-vacuum state, the corresponding pseudo-vacuum states should differ. However, these states are equal, as follows from

$$\text{(65)}$$

In the first equality we have used Eq. (62) and in the second equality we used the braiding relation (20). Similar manipulations can be used to show that unitary transformations only corresponding to permutations within the 'uncontracted' inhomogeneities $\epsilon_{P(1)} \dots \epsilon_{P(n<m)}$ do not lead to additional vacuum states for $|\emptyset\rangle_{m,n}$.

**Exchanging contracted sites.** A similar argument can be used when exchanging contracted inhomogeneities $\epsilon_{P(n+1)} \dots \epsilon_{P(m)}$, which we will now illustrate on the vacuum state with $n = 1$. For the $m = 3$ transfer matrix this results in eigenvalues depending on a single inhomogeneity $\epsilon_{P(1)}$. Suppose we want the states with eigenvalue depending on $\epsilon_3$. There are again two possible permutations returning the desired result

$$(1,2,3) \xrightarrow{(23)} (1,3,2) \xrightarrow{(13)} (3,1,2), \tag{66}$$

$$(1,2,3) \xrightarrow{(12)} (2,1,3) \xrightarrow{(13)} (2,3,1) \xrightarrow{(23)} (3,2,1), \tag{67}$$

now only differing in the two elements corresponding to the contracted inhomogeneities. The equality of the vacuum states now follows from

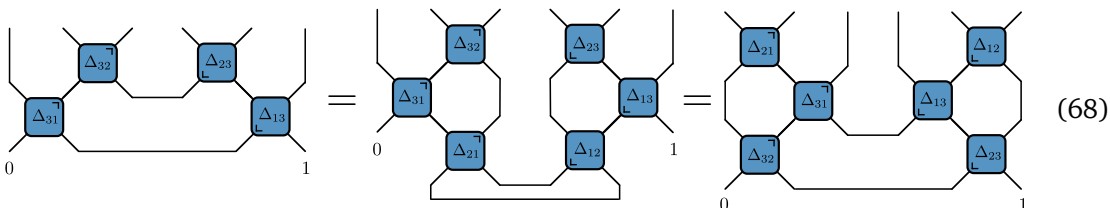

$$\tag{68}$$

where the first equality follows from Eq. (61) and the second equality again follows from the braiding relation (20).

**Completeness.** The arguments above can immediately be extended to general transfer matrices. Introducing inhomogeneities, any selection of $n$ out of $m$ inhomogeneities returns a distinct vacuum state. The Bethe states obtained by acting on this vacuum state follow by solving the Bethe equations for the spin-1 chain with $n$ sites and the selected inhomogeneities $\{\epsilon_{P(1)}, \dots, \epsilon_{P(n)}\}$. Both the Bethe equations and eigenvalues depend on the choice of inhomogeneities, such that different selections generally lead to different eigenvalues. From the completeness of the Bethe equations for the inhomogeneous spin-1 chain, any such vacuum state can be used to construct $3^m$ states. Combining the exponential number of states for a given vacuum state with the combinatorial number of vacuum states then results in

$$\sum_{n=0}^{m} \binom{m}{n} 3^n = 4^m, \tag{69}$$

the expected number of states for the transfer matrix. Assuming all eigenvalues to be different, up to global $SU(2)$ symmetry (see Appendix A.2), the inhomogeneous transfer matrix is diagonalizable and all states can be written as a Bethe ansatz. This construction is lost in the homogeneous limit, since then all homogeneities coalesce and all unitaries $U_P$ reduce to the identity. As such, states with distinct eigenvalues in the inhomogeneous case will coalesce in the homogeneous limit, leading to a transfer matrix that is no longer diagonalizable and necessitating nontrivial Jordan blocks in the Jordan decomposition. In Appendix A.1 it is shown how generalized eigenstates can be obtained from the Bethe ansatz.

**Connection to spin-1 models.** The introduction of inhomogeneities also allows for an explicit connection with the spin-1 construction, making clear why the obtained Bethe equations correspond to the ones for the integrable spin-1 chain with $m$ sites. Assuming all inhomogeneities to be distinct, the inhomogeneities can be reordered through a unitary transformation

to show that the monodromy matrix is unitarily equivalent to

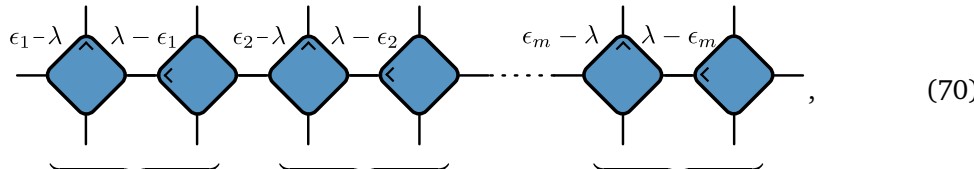

$$(70)$$

where all inhomogeneities have been paired up. This can be seen as a monodromy matrix built out of inhomogeneous building blocks $\tilde{R}(\lambda)$ acting on two copies of the local Hilbert space (and the auxiliary space)

$$\tilde{R}(\lambda) = \quad (71)$$

This doubled local space can now be interpreted as the space of all operators acting on the local Hilbert space. We can evaluate the action of $\tilde{R}(\lambda)$ on the Pauli matrices by evaluating

$$a \quad b = \frac{1}{1+\lambda^2}\left(\mathrm{Tr}[\sigma_\alpha](\sigma_\beta)_{ab} - i\lambda[\sigma_\alpha,\sigma_\beta]_{ab} + \lambda^2\delta_{ab}\mathrm{Tr}[\sigma^\alpha\sigma^\beta]\right), \quad (72)$$

where we have only used the definition of the original $\check{R}$-matrix (16) to evaluate these matrix elements. For $\sigma_\beta = \mathbb{1}$ we find the expression for the unitarity, where the identity operator is a trivial eigenstate of $\tilde{R}(\lambda)$. When taking both $\alpha, \beta \in \{x, y, z\}$, the commutator of the Pauli matrices can be explicitly evaluated to return

$$a \quad b = \frac{2\lambda}{1+\lambda^2}\left(\lambda\delta_{\alpha\beta}\delta_{ab} + \sum_\gamma \epsilon_{\alpha\beta\gamma}(\sigma_\gamma)_{ab}\right) \quad (73)$$

$$= \frac{2\lambda}{1+\lambda^2}\left(\lambda\delta_{ab}\mathbb{1}_{\alpha\beta} + i\sum_\gamma (S^\gamma)_{\alpha\beta}(\sigma_\gamma)_{ab}\right), \quad (74)$$

where $\mathbb{1}$ and $S^\gamma$ act on the space of Pauli operators and $S^\gamma, \gamma = x, y, z$ are the spin-1 matrices satisfying $S^\gamma_{\alpha\beta} = -i\epsilon_{\alpha\beta\gamma}$. These are exactly the matrix elements of the Lax operator for the spin-1 chain [53, 55]. In other words, we can interpret the full transfer matrix as a transfer matrix constructed out of $m$ operators $\tilde{R}(\lambda)$ acting on a four-dimensional local Hilbert space, identifying this Hilbert space with the space of operators $\{\sigma_\alpha, \alpha = 0, x, y, z\} \in \mathbb{C}^{2\times2}$, and $\tilde{R}(\lambda)$ acts trivially on the identity ($\alpha = 0$) and acts as the spin-1 Lax matrix on the Pauli matrices ($\alpha = x, y, z$). This can also be seen by combining the local unitary transformation introduced in Sec. 2.2 with the fusion procedure of Ref. [57].

The eigenvalue equation for this transfer matrix returns the expected spin-1 Bethe equations, and after reordering the inhomogeneities in both the transfer matrix and Bethe state the homogeneous limit can be taken. Note that a similar mapping between spin-1/2 operator dynamics and a non-Hermitian spin-1 integrable model was also observed in Ref. [58]

## 2.5 Eigenvalues

The generalized eigenvalues depend on both the choice of spectral parameter and the size of the transfer matrix. While the transfer matrix can be numerically diagonalized, the exponential growth of the Hilbert space with $m$ limits calculations to small sizes. The Bethe equations Eqs. (42) are known in the literature and various strategies have been developed in order to efficiently solve these nonlinear equations [55,56].

We here focus on specific properties of the eigenspectrum of the transfer matrix $\tau_m(\lambda)/2$ for the unitary circuit in the homogeneous limit, where the spectrum for $m = 1,\ldots,4$ is given in Fig. 1. Here we have reintroduced a normalization prefactor $1/2$ in order to be consistent with Section 1.1. Several properties can immediately be observed. We denote the eigenvalues as $t_m(\lambda)$, suppressing the dependence on the rapidities, and all eigenvalues can be seen to satisfy $|t_m(\lambda)| \le 1$. At each choice of spectral parameter a single eigenvalue $t_m(\lambda) = 1$ is guaranteed to be present with corresponding eigenvector $|\emptyset\rangle_{m,m}$. At $\lambda = 0$ the transfer matrix is a projector, with a nonzero single eigenvalue one and all other eigenvalues zero, whereas for $\lambda \to \infty$ the transfer matrix reduces to the identity and all eigenvalues equal one. Furthermore, the previously mentioned nesting of the eigenspectrum is clear, where the eigenspectrum of $\tau_m(\lambda)$ contains the eigenspectrum of the transfer matrices $\tau_{n<m}(\lambda)$. These nested eigenvalues are highlighted red.

Furthermore, it can be observed that there are exact degeneracies present in the spectrum, persisting at all values of the spectral paramterer. This is a direct consequence of the total $SU(2)$ symmetry of the system, such that all eigenstates of $\tau_m(\lambda)$ can be characterized by a total spin $\tilde{S}^2$ and the eigenvalues are independent of the corresponding spin projection $\tilde{S}^z$. This symmetry is explicitly shown in Appendix A.2, containing an explicit derivation of the symmetry generators. We here only note that the pseudo-vacuum states are extremal weight states, satisfying

$$\tilde{S}^z |\emptyset\rangle_{m,n} = -n |\emptyset\rangle_{m,n} , \qquad \tilde{S}^2 |\emptyset\rangle_{m,n} = n(n+1) |\emptyset\rangle_{m,n} . \tag{75}$$

The full spectrum for $m = 1$ and $m = 2$ can be analytically obtained. For $m = 1$ there are two eigenvalues, necessarily corresponding to the two vacuum states $|\emptyset\rangle_{1,1}$ and $|\emptyset\rangle_{1,0}$. The first state returns the trivial eigenvalue one with $\tilde{S} = 0$, whereas the eigenvalues for the second

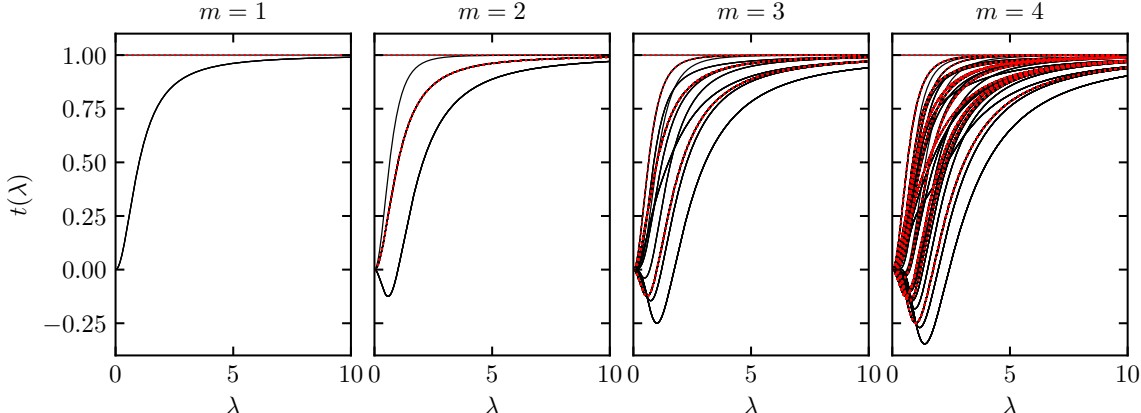

Figure 1: Eigenvalues $t_m(\lambda)$ of the transfer matrices $T_m(\lambda)$ at different sizes. Black lines denote the eigenspectrum at fixed $m$, dotted red lines are overlaid on eigenvalues obtained from smaller transfer matrices, highlighting the nesting of the spectrum.

state immediately follow as

$$t(\lambda) = \frac{\lambda^2}{1+\lambda^2}.$$ (76)

The corresponding eigenstate has total spin $\tilde{S} = 1$, leading to a threefold degeneracy due to the total spin symmetry, such that the full spectrum is obtained and $\tau_1(\lambda)$ is generally diagonalizable.

For $m = 2$ two additional eigenvalues can be identified. The corresponding states are given by the vacuum state for $m = 2$, now with eigenvalue

$$t(\lambda) = \lambda^2 \frac{(\lambda^2 - 1)}{(\lambda^2 + 1)^2}.$$ (77)

This state has total spin 2 and is five-fold degenerate.

The nontrivial state can be obtained from the Bethe equations (42) for $m = 2$ and $N = 2$, which can be analytically solved to return two rapidities $\pm i/\sqrt{3}$. Plugging this into the eigenvalue (48) returns a single non-degenerate eigenvalue

$$t(\lambda|\{\pm i/\sqrt{3}\}) = \lambda^2 \frac{(\lambda^2 + 2)}{(\lambda^2 + 1)^2}.$$ (78)

The eigenvalue associated with the pseudo-vacuum $|\emptyset\rangle_{1,1}$ now corresponds to three three-dimensional Jordan blocks rather than the expected two. This can be understood by noting that a single zero rapidity solves the Bethe equations (42) for $m = 2$ with $N = 1$. Introducing inhomogeneities $\epsilon_{1,2}$, this single rapidity can be analytically found as $(\epsilon_1 + \epsilon_2)/2$. In the inhomogeneous case, this Jordan block hence returns three three-fold degenerate states with $\tilde{S} = 1$ and eigenvalues

$$\frac{(\lambda - \epsilon_1)^2}{1 + (\lambda - \epsilon_1)^2}, \qquad \frac{(\lambda - \epsilon_2)^2}{1 + (\lambda - \epsilon_2)^2}, \qquad \frac{(\lambda - \epsilon_1)(\lambda - \epsilon_2)(1 + \epsilon_1\epsilon_2 - (\epsilon_1 + \epsilon_2)\lambda + \lambda^2)}{(1 + (\lambda - \epsilon_1)^2)(1 + (\lambda - \epsilon_2)^2)}.$$ (79)

Taking these together then returns the expected dimension of the Hilbert space $1 + 5 + 1 + 9 = 16$.

In fact, the spectrum of $m = 3$ transfer matrix and its Jordan normal form can also be obtained analytically even though the matrix has dimension $64 \times 64$. The results are reproduced in Table 1. We confirm the nesting of the four $m = 2$ eigenvalues within the $m = 3$ spectrum. While the spin 2 state in Eq. (77) was an exact eigenstate in the $m = 2$ transfer matrix, it now appears in a nontrivial Jordan block at $m = 3$, splitting into five $3 \times 3$ Jordan blocks. None of the seven new $m = 3$ eigenvalues have nontrivial Jordan blocks. We also observe that the seven additional eigenvalues of $\tau_3(\lambda)$ have denominators of the form $(1 + \lambda^2)^3$, matching the pattern of $(1 + \lambda^2)^2$ denominators of the $\tau_2(\lambda)$ eigenvalues and $(1 + \lambda^2)$ denominators of the $\tau_1(\lambda)$ eigenvalues.

We note that a general method for constructing the transfer matrix within a generalized eigenspace is given in Ref. [59] by Mukhin, Tarasov, and Varchenko. The Bethe equations are in bijection with a set of polynomial equations, which can be considered as algebraic equations in the coefficients of these polynomials, and each solution of those algebraic equations has an associated local algebra that is in turn isomorphic to the blocks within a generalized eigenspace. However, these blocks may then have a nontrivial Jordan decomposition, such that the size of the Jordan blocks does not necessarily correspond to the dimension of this algebra. As one example, in Table 1 the eigenvalue $\lambda^2/(1+\lambda^2)$ is associated with a six-dimensional local algebra within each fixed symmetry sector, and each $6 \times 6$ block splits into a $1 \times 1$ and $5 \times 5$ Jordan block. We refer the reader to Ref. [59] for more details and only note that for all presented eigenvalues the dimensions of the Jordan blocks correspond to those predicted by the approach of [59].

Table 1: Jordan Decomposition of $\tau_3(\lambda)$. We find 11 different eigenvalues $t(\lambda)$ with degeneracies and Jordan block dimensions given above. For instance, the $\lambda^2/(1+\lambda^2)$ eigenvalue is 18-fold degenerate and splits into three $5 \times 5$ and three $1 \times 1$ Jordan blocks for all finite nonzero $\lambda$.

| $t(\lambda)$ | $1$ | $\frac{\lambda^2}{1+\lambda^2}$ | $\lambda^2 \frac{\lambda^2+2}{(1+\lambda^2)^2}$ | $\lambda^2 \frac{\lambda^2-1}{(1+\lambda^2)^2}$ |
|---|---|---|---|---|
| Degen. | $1$ | $18$ | $3$ | $15$ |
| Blocks | $1 \times 1$ | $5 \times 5 + 1 \times 1$ | $3 \times 3$ | $3 \times 3$ |
| $t(\lambda)$ | $\lambda^3 \frac{\lambda^3+2\lambda+\sqrt{3}}{(1+\lambda^2)^3}$ | $\lambda^4 \frac{\lambda^2-3}{(1+\lambda^2)^3}$ | $\lambda^4 \frac{\lambda^2+3}{(1+\lambda^2)^3}$ | $\lambda^4 \frac{\lambda^2+2}{(1+\lambda^2)^3}$ |
| Degen. | $3$ | $7$ | $1$ | $3$ |
| Blocks | $1 \times 1$ | $1 \times 1$ | $1 \times 1$ | $1 \times 1$ |
| $t(\lambda)$ | $\lambda^3 \frac{\lambda^3-\sqrt{3}}{(1+\lambda^2)^3}$ | $\lambda^3 \frac{\lambda^3+\sqrt{3}}{(1+\lambda^2)^3}$ | $\lambda^3 \frac{\lambda^3+2\lambda-\sqrt{3}}{(1+\lambda^2)^3}$ | |
| Degen. | $5$ | $5$ | $3$ | |
| Blocks | $1 \times 1$ | $1 \times 1$ | $1 \times 1$ | |

## 3 Out-of-time-order correlators

Out-of-time-order correlators (OTOCs) can similarly be calculated, and we here illustrate how the above construction gives rise to a different family of commuting transfer matrices. We will consider OTOCs of the form

$$
\begin{aligned}
C_{\alpha\beta}(x,t) &= \langle \sigma_\alpha(0,0)\sigma_\beta(x,t)\sigma_\alpha(0,0)\sigma_\beta(x,t) \rangle \\
&= \langle \sigma_\alpha(0)\mathcal{U}(t)\sigma_\beta(x)\mathcal{U}^\dagger(t)\sigma_\alpha(0)\mathcal{U}(t)\sigma_\beta(x)\mathcal{U}^\dagger(t) \rangle .
\end{aligned}
\tag{80}
$$

In a similar way as for the time-ordered correlators of Eq. (4), the evaluation of the OTOC can be reduced to the contraction of a finite diagram (as also detailed in Ref. [26]),

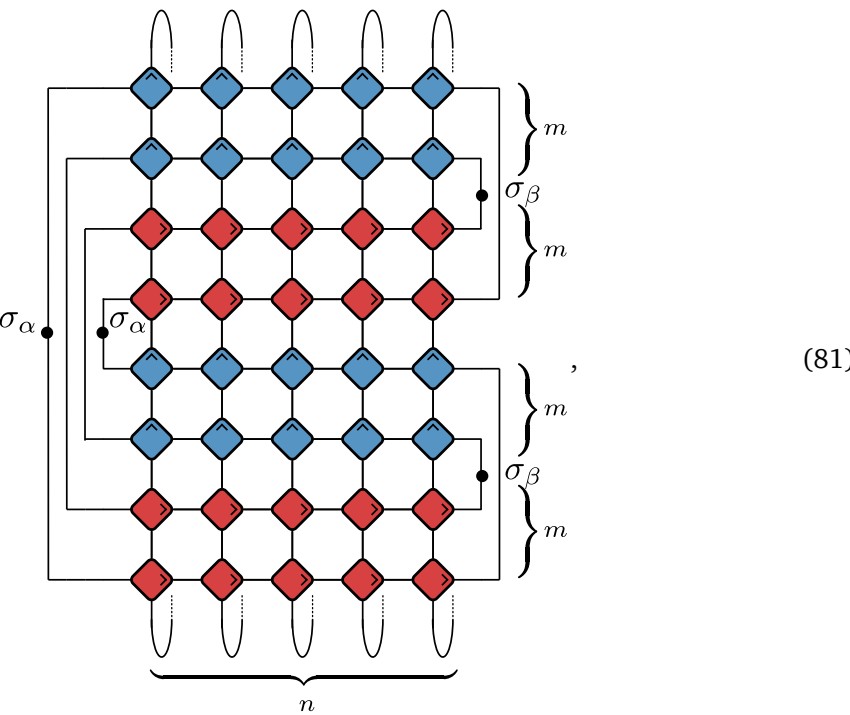

$$\tag{81}$$

where a monodromy matrix and resulting transfer matrix can again be identified as

$$
T_m(\lambda) = \qquad\qquad\qquad\qquad\qquad\qquad\qquad\qquad\qquad . \tag{82}
$$

These now exhibit a four-component rather than a two-component structure, reflecting the appearance of two copies of both $\mathcal{U}(t)$ and $\mathcal{U}^\dagger(t)$ in the OTOC. Since the following results are completely analogous to the result for the correlation functions, we here only mention the main results. These monodromy matrices satisfy the same RTT relation as the monodromy matrices for the correlation functions, such that the transfer matrices again necessarily commute. The Bethe ansatz construction also holds, where a set of pseudo-vacuum states can now be found as $|\emptyset\rangle_{m,n} \otimes |\emptyset\rangle_{m,l}$, $n, l = 0 \dots m$. The eigenvalues of the resulting Bethe states are given by

$$
t(\mu|\lambda_1 \dots \lambda_N) = \left(\frac{i\mu}{1+i\mu}\right)^{n+l} \prod_{k=1}^{N}\left(1 + \frac{i}{\lambda_k - \mu}\right) + \left(\frac{-i\mu}{1-i\mu}\right)^{n+l} \prod_{k=1}^{N}\left(1 - \frac{i}{\lambda_k - \mu}\right), \tag{83}
$$

with the Bethe equations now the equations for the spin-1 chain with $n+l$ sites,

$$
\left(\frac{\lambda_j - i}{\lambda_j + i}\right)^{n+l} = \prod_{k \neq j}^{N} \frac{\lambda_k - \lambda_j + i}{\lambda_k - \lambda_j - i}. \tag{84}
$$

Using the same argument as for the correlation functions, in the inhomogeneous model a complete set of eigenstates is obtained, since the total number of vacuum states and resulting Bethe states is given by

$$
\sum_{n=0}^{m}\sum_{l=0}^{m} \binom{m}{n}\binom{m}{l} 3^{k+n} = 16^m, \tag{85}
$$

returning the total dimension of the Hilbert space.

## 4  Conclusion

In this work, we showed how correlation functions in unitary circuits can be calculated using a transfer matrix approach, and analyzed the properties of the resulting transfer matrices in the case where the underlying gates are $\check{R}$-matrices satisfying the Yang-Baxter equation. The transfer matrices determine the correlations in the limit of infinite system size, but their dimension is fully determined by the distance from the causal light cone. These matrices exhibit a two-component structure, refelecting the necessary forward and backward propagation in time, and are shown to commute at different values of the spectral parameter – despite the fact that different spectral parameters here define different circuits with different conservation laws. The specific two-component structure leads to an additional structure as compared to one-component transfer matrices, and a nesting of the eigenspectrum is observed. In homogeneous systems it was shown that the transfer matrix was no longer diagonalizable, but its eigenstates could still be obtained as Bethe states, with an extensive number of vacuum states. Introducing inhomogeneities, the resulting transfer matrices are diagonalizable, with the Bethe ansatz requiring a combinatorial number of vacuum states. A similar construction was shown to hold for the calculation of out-of-time-order correlations.

At long times, the correlations in integrable circuits with a nonabelian symmetry such as $SU(2)$ are expected to return Kardar-Parisi-Zhang (KPZ) dynamics [40, 42–45]. Unfortunately, the complicated Jordan block structure of the transfer matrix prevents a direct connection

between the presented eigenstates and the long-time dynamics. Here, we only comment on how the involved transfer matrices prevent a straightforward exponential decay at long times – the decay of correlations for fixed $n$ is not simply determined by the dominant eigenvalue. The underlying mechanism differs in homogeneous and inhomogeneous chains. In the homogeneous case, it is the appearance of Jordan blocks, increasing in size as the dimension of the transfer matrix increases. The presence of Jordan blocks results in exponential decay with a polynomial prefactor depending on the choice of spectral parameter, further complicating immediate analysis. In the inhomogeneous case the transfer matrix is diagonalizable, but suffers from diverging overlaps of the left and right eigenvectors with the boundary states, due to the nearby exceptional point. Such a behaviour has also recently been observed in unitary circuits [60] and open systems, where the dominant eigenvalue similarly is not guaranteed to determine the long-time dynamics [61].

## Acknowledgements

We gratefully acknowledge support from EPSRC Grant No. EP/P034616/1. JHA is supported by a Marshall Scholarship. We are grateful to V. Tarasov for various clarifying remarks and useful comments on the manuscript.

## A  Appendices

### A.1  The homogeneous limit

Following the arguments of the main text, the fully inhomogeneous transfer matrix is diagonalizable. In the homogeneous limit all inhomogeneities vanish (or are equal), and $\Delta_{ij} = 0$. In this limit the unitary transformation reordering the inhomogeneities reduces to the identity, such that the different eigenstates corresponding to reordered inhomogeneities and contracted vacuum states coalesce. This is exactly what happens in exceptional points, where a non-hermitian operator is no longer diagonalizable but rather requires a Jordan decomposition including nontrivial Jordan blocks. Such Jordan blocks have previously been observed in integrable systems [62, 63], where generalized eigenvectors have also been obtained in some cases.

It is important to note that the commutativity of transfer matrices at different values of the spectral parameters does not imply that they have a common Jordan decomposition. Furthermore, the Jordan decomposition does not depend continuously on the spectral parameter. This is already made clear from the limit where the spectral parameter $\mu \to \infty$: in this limit the individual gates reduce to swap gates and the transfer matrix reduces to the identity, which has a trivial Jordan decomposition. Away from this limit, Jordan blocks appear.

The explicit construction of Jordan blocks requires the construction of generalized eigenvectors. We will illustrate how these can be obtained from the inhomogeneous case by considering the limit where two inhomogeneities become equal. For ease of notation and to fix ideas, all graphics will be restricted to $m = 3$ and $n = 2$, but the construction holds more generally. We denote

$$T(\Delta) = \quad , \tag{86}$$

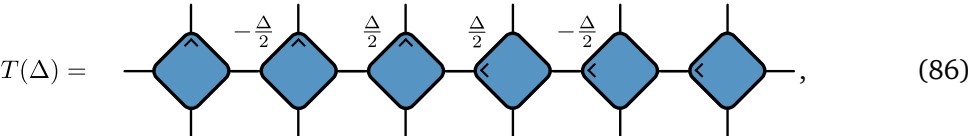

where two inhomogeneities $\pm \Delta/2$ have been introduced, such that the two sites are homogeneous in the limit $\Delta \to 0$. Other inhomogeneities might be present in the transfer matrix,

but those will not be important for the following derivation. A unitary transformation can be introduced



$$U(\Delta) = \qquad \qquad \qquad , \qquad (87)$$

where the dependence on $\Delta$ has been chosen such that $T(-\Delta) = U^\dagger(\Delta)T(\Delta)U(\Delta)$, leading to

$$\partial_\Delta T(0) = \frac{i}{2}[S, T(0)], \qquad \text{with} \qquad \partial_\Delta U(0) = iS. \qquad (88)$$

At every value of the inhomogeneities we can find an eigenstate

$$T(\Delta)|\psi(\Delta)\rangle = t(\Delta)|\psi(\Delta)\rangle. \qquad (89)$$

Crucially, the eigenvalue $t(\Delta)$ depends on the difference in inhomogeneities in a nontrivial way since $|\psi(\Delta)\rangle \neq U^\dagger(\Delta)|\psi(0)\rangle$ (which would be the case if we were exchanging homogeneities that are either both contracted or both uncontracted). This state is a Bethe state depending on $\Delta$ both indirectly (through the single inhomogeneity $\epsilon + \Delta/2$ in the Bethe equations) and directly (through the definition of $B(\lambda)$).

Taking the derivative of this equation and making use of Eq. (88), we find the generalized eigenvalue equation

$$T(0)\left[|\partial_\Delta \psi(0)\rangle - \frac{i}{2}S|\psi(0)\rangle\right] = t(0)\left[|\partial_\Delta \psi(0)\rangle - \frac{i}{2}S|\psi(0)\rangle\right] + \partial_\Delta t(0)|\psi(0)\rangle, \qquad (90)$$

such that generalized eigenvectors are given by

$$|\partial_\Delta \psi(0)\rangle - \frac{i}{2}S|\psi(0)\rangle = \partial_\Delta\left[U^\dagger(\Delta/2)|\psi(\Delta)\rangle\right]_{\Delta=0}. \qquad (91)$$

This can be recast by making use of the explicit Bethe ansatz construction,

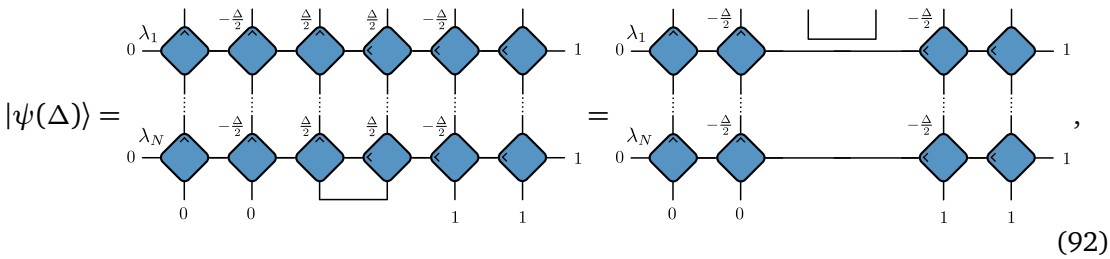

$$|\psi(\Delta)\rangle = \qquad \qquad = \qquad \qquad , \qquad (92)$$

consisting of $N$ rows of $B(\lambda)$ operators, and where we have made explicit the dependence on the inhomogeneity in the raising operators. The action of the unitary then returns

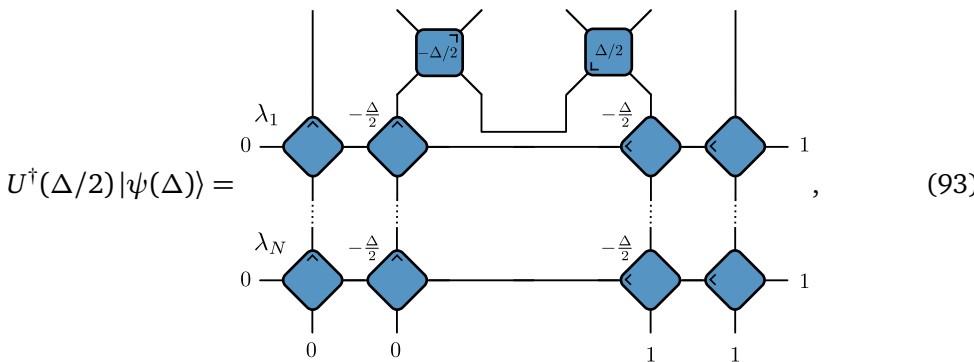

$$U^\dagger(\Delta/2)|\psi(\Delta)\rangle = \qquad \qquad , \qquad (93)$$

which can be simplified by reintroducing contracted unitaries with zero inhomogeneity, using the braiding relation to exchange two pairs of columns, leading to

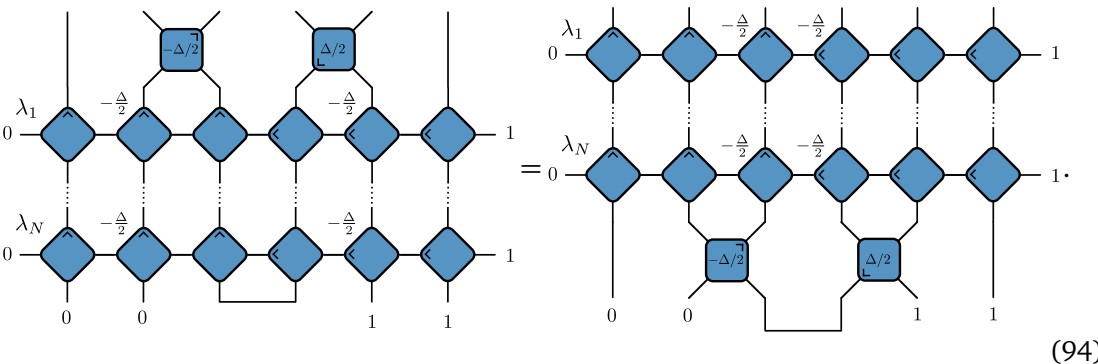

$$(94)$$

This consists of a set of generalized raising operators acting on a generalized vacuum. The Bethe vector depends on the $N$ Bethe roots $\{\lambda_1 \ldots \lambda_N\}$, obtained by solving the Bethe equations with nonzero inhomogeneity $-\Delta/2$, but evaluated using the creation operators $B(\lambda)$ where the inhomogeneity has been moved. Evaluating the derivative, this returns a generalized eigenvector

$$\prod_{k=1}^{N} B(\lambda_k) |\partial_\Delta \emptyset(0)\rangle + \sum_{k=1}^{N} \left( \prod_{l \neq k} B(\lambda_l) \right) d_\Delta B(\lambda_k) |\emptyset_0\rangle \,. \tag{95}$$

The explicit dependence of $B$ on $\Delta$ can be removed since the action of $B(\lambda)$ on $|\emptyset_0\rangle$ contracts the central unitaries, such that any dependence on the parametrization of these unitaries drops out.

$$|\lambda_1 \ldots \lambda_N; \partial_\Delta \lambda_1 \ldots \partial_\Delta \lambda_N\rangle = \prod_{k=1}^{N} B(\lambda_k) |\partial_\Delta \emptyset(0)\rangle_{m,1} + \sum_{k=1}^{N} \partial_\Delta \lambda_k \cdot \partial_\lambda B(\lambda_k) \left( \prod_{j \neq k}^{N} B(\lambda_j) \right) |\emptyset\rangle_{m,1} \,, \tag{96}$$

satisfying the generalized eigenvalue equation

$$[T(\lambda) - t(\lambda)] |\lambda_1 \ldots \lambda_N; \partial_\Delta \lambda_1 \ldots \partial_\Delta \lambda_N\rangle = \partial_\Delta t(\lambda) |\lambda_1 \ldots \lambda_N\rangle \,. \tag{97}$$

This method can be extended to higher-order generalized eigenvectors since the only requirement is Eq. (88), but we leave an explicit construction to future work. Alternatively, it seems likely that the ABA construction using generalized commutation relations can be extended to reproduce the above generalized eigenvectors.

## A.2 $SU(2)$ symmetry

The transfer matrix exhibits a total $SU(2)$ symmetry, both in the homogeneous and inhomogeneous case. The generators of this symmetry can be obtained from

$$T_m(\lambda \to \infty) = \begin{bmatrix} 1 & 0 \\ 0 & 1 \end{bmatrix} - \frac{i}{\lambda} \begin{bmatrix} \tilde{S}^z & \tilde{S}^+ \\ \tilde{S}^- & -\tilde{S}^z \end{bmatrix} + \mathcal{O}(\lambda^{-2}), \tag{98}$$

with $\tilde{S}^\pm = \tilde{S}^x \pm i\tilde{S}^y$, with the local basis choice $|0\rangle \to |\downarrow\rangle$ and $|1\rangle \to |\uparrow\rangle$ and

$$\tilde{S}^z = \sum_{i=1}^{m} S_i^z - \sum_{i=m+1}^{2m} S_i^z, \quad \tilde{S}^x = -\sum_{i=1}^{m} S_i^x + \sum_{i=m+1}^{2m} S_i^x, \quad \tilde{S}^y = -\sum_{i=1}^{m} S_i^y - \sum_{i=m+1}^{2m} S_i^y. \tag{99}$$

The transfer matrix $\tau_m(\lambda)$ commutes with all $\tilde{S}^\alpha$ and $\tilde{S}^2 = \left(\tilde{S}^x\right)^2 + \left(\tilde{S}^y\right)^2 + \left(\tilde{S}^z\right)^2$, as follows from the generalized commutation relations in the limit $\lambda \to \infty$. The quantum numbers of the vacuum states can be found as

$$\tilde{S}^z \, |\emptyset\rangle_{m,n} = -n \, |\emptyset\rangle_{m,n} \,, \qquad \tilde{S}^2 \, |\emptyset\rangle_{m,n} = n(n+1) \, |\emptyset\rangle_{m,n} \,. \tag{100}$$

These are again minimal weight states, and the Bethe raising operators increase $\tilde{S}^z$ by one.

In the calculation of one-site correlation functions, only Bethe states with $\tilde{S} = 1$ will be relevant since it can easily be checked that the boundary vectors (11) and (10) have a well-defined total spin $\tilde{S} = 1$.

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
