# Peer review of "Correlations and commuting transfer matrices in integrable unitary circuits"

_SciPost Physics, doi:SciPost Phys. 12, 007 (2022)_

## Round 2 · Referee Report · Anonymous (Referee 1) · 2021-8-9

Strengths

1) It deals with an interesting and timely problem, namely computation of correlators in unitary circuits 2) Uses advanced techniques in integrable systems 3) Obtains interesting and novel results 4) It is very well written 5) contains an updated list of references.

Weaknesses

1) contains some technical parts that could be moved to appendices, making the paper more readable.

Report

This paper proposes an interesting method to compute infinite temperature correlation functions between one-site operators in integrable quantum circuits. The latter are given by a trotterization of the evolution operator of an integrable
Hamiltonian, in this case the spin 1/2 Heisenberg Hamiltonian.
The authors formulate the problem using a non-standard transfer matrix with a two-component structure, that depends on a spectral parameter. They show that transfer matrices, with different spectral parameters, commute thanks to the Yang-Baxter equation satisfied by the R matrix of the XXX model. They use the algebraic Bethe ansatz (ABA) to diagonalize the transfer matrix that,
quite interestingly, turns out to coincide with the Bethe ansatz equation for a spin 1 XXX model. The diagonalization has also some subtleties that lead the authors to consider the inhomogeneous version of the model that enable them to show the completeness of the ABA and other subtleties like the Jordan like structure of the transfer matrix in the homogenous case. In some simple cases the spectrum of the transfer matrix is given analytically showing the nesting property and the Jordan decomposition in the homogeneous case. The paper is very welll written and contains highly interesting results.

Requested changes

1) In the footnote in page 9 it is said that $\tau_m(\lambda)$ is the monodromy matrix and $T_m(\lambda)$ is the transfer matrix. Please correct, the notation is the opposite.

2) The standard RTT equation is formulated in terms of the "universal" R matrix that satisfies the Yang-Baxter eq. in the form

R12(u) R13(u+v) R23(v) = R23(v) R13(u+v) R12(u)

that differs from the one given in eq. (19) that corresponds to "braiding" R matrices. A comment on this issue will be welcome.

3) Section 2.4 is devoted to the inhomogenous model is interesting but a bit technical. I would suggest the authors to move some parts to an appendix to make the manuscript more readable.

4) In page 17 it is explained why the Bethe eq. for the roots is the same as those for the spin 1 - XXX model of Babujian and Taktajan. Is there a relation between this derivation and the one in reference Kulish, Resehtikhin and Sklyanin, Lett. Matt. Phys. 5, 393 (1981) where the R matrices for higher spins are derived using the spin 1/2 R -matrix?

5) I would be helpful for the reader to provide the form of the BAE for a generic spin S, so that the statement that eq.(42) is the case S=1, will be readily understood.

  • validity: top
  • significance: high
  • originality: high
  • clarity: high
  • formatting: perfect
  • grammar: perfect

Author:  Pieter W. Claeys  on 2021-10-08  [id 1824]

(in reply to Report 1 on 2021-08-09)

We would like to thank the Referee for their clear report with a detailed reading of our manuscript and useful comments and pointers to the literature. As mentioned in our resubmission, we have attempted to address all requested changes and hope this makes our paper acceptable for publication in SciPost Physics Core.

For the authors,
Pieter W. Claeys

---

## Round 2 · Referee Report · Anonymous (Referee 2) · 2021-8-19

Strengths

  • computation of dynamical correlation functions in integrable lattice systems is addressed directly in terms fo an appropriate transfer matrix
  • the technique has a clear diagrammatic representation in terms of basic objects of integrability

Weaknesses

  • it is a unfortunate that explicit results on (asymptotic) of correlation functions are still out of reach, even in simple special cases
  • it would be good to see some calculations in more explicit form

Report

The paper proposes an explicit expression for 2-point (as well as 4-point, in case of OTOCs) spatio-temporal correlation functions of local observables in terms of a specific inhomogeneous and non-unitary transfer matrix. This approach works particularly nicely for models which are naturally written as integrable brickwork circuits with the elementary two-site gate obeying the braid form of Yang-Baxter equation. Clearly, the approach can be extended to integrable Hamiltonians by means of the Trotter formula.

It is nice to see an alternative, fresh approach to computation of dynamical correlations in integrable systems which circumvents the use of cumbersome form factor expansions. Still, many technical difficulties remain which prevent one from obtaining explicit asymptotic results, say, addressing the anomalous (super-diffusive) KPZ spin transport in trotterised XXX model. These difficulties, mainly related to non-diagonalizability and non-unitarity of the transfer matrix, are clearly discussed and explained. Appropriate Bethe equations for the correlation transfer matrix spectrum are spelled out and related to spin-1 integrable chains.

The paper is clearly written. Several technical derivations are nicely presented using diagrammatic technique. Despite lacking really useful results, I think it will be a valuable addition to the literature and could perhaps stimulate further development of this important problem. Therefore, I recommend it for publication in SciPost.

Requested changes

  • For the sake of compatibility with the literature, it is perhaps worth mentioning at some place that what authors denote as $R$-matrix is usually referred to as matrix $\check R = P R$.

  • Misleading sentence in the conclusion: KPZ universality scaling of correlation functions is expected only in integrable systems with nonabelian symmetries, e.g. in gapped XXZ model one finds diffusive transport. This has to be stressed correctly.

  • Optional: I wander if the formulation of the correlation function transfer matrix would looks simpler if the authors would use the "folded representation" from the beginning (?)

  • validity: high
  • significance: high
  • originality: high
  • clarity: high
  • formatting: perfect
  • grammar: perfect

Author:  Pieter W. Claeys  on 2021-10-08  [id 1823]

(in reply to Report 2 on 2021-08-19)

We would like to thank the Referee for their clear report with a detailed reading of our manuscript and useful comments. As mentioned in our resubmission, we have attempted to address all requested changes and hope this makes our paper acceptable for publication in SciPost Physics Core.

For the authors,
Pieter W. Claeys

---

## Round 3 · Referee Report · Anonymous (Referee 2) · 2021-10-14

Report

I believe the authors have successfully addressed all remarks and improved the presentation. I am happy to recommend the publication of the mansucript in SciPost Physics in the present form.

---

## Round 3 · Author Response

We would like to thank both referees for their careful reading of our manuscript, their useful suggestions, and their positive assessment. In the revised version we have addressed all requested changes and made some minor revisions which we hope makes our paper acceptable for publication in SciPost Physics Core.

For the authors,
Pieter W. Claeys

---

## Round 3 · List of Changes

The main changes in the manuscript can be found below.

  • The notation of transfer matrix and monodromy matrix has been made consistent with the literature throughout the manuscript.
  • We have made clear that we are working with the $\check{R}$-matrix rather than the $R$-matrix, and explicitly mentioned the connection between the two when introducing the braiding relation in Eq. (19).
  • The connection with the fusion procedure of Kulish, Resehtikhin and Sklyanin, Lett. Matt. Phys. 5, 393 (1981) has been explicitly mentioned at the end of Section 2.4.
  • The BAE for arbitrary spin s have been given in Eq. (43), making it easy to identify our obtained BAE [Eq. (42)] as those for the integrable spin-1 chain.
  • In the conclusion we now explicitly mention the importance of nonabelian symmetries for KPZ.
  • In Table 1 the dimensions of the Jordan blocks associated with the eigenvalue $\lambda^2/(1+\lambda^2)$ have been corrected: rather than three 6x6 blocks, this should have been three 5x5 and three 1x1 blocks. At the end of Section 2.5 we also provide a reference and short discussion of the work of the added Ref. [59], which allowed us to spot this mistake and presents a useful way of constructing the matrix elements of the transfer matrix in the generalized eigenbasis, but unfortunately does not allow for a direct calculation of the dimension of the Jordan blocks in general.
  • Minor typographical changes have been made throughout.

We have chosen to keep Section 2.4 on the inhomogeneous model as is. We agree that the section is technical, but we also believe that the results in there are important for establishing the completeness of the Bethe ansatz in the inhomogeneous case, as well as further clarifying in which ways the presented two-component transfer matrices differ from the one-component transfer matrices usually studied. Parts of this section had also already been moved to Appendix before submission. To accommodate this remark we now mention at the beginning of Section 2.4 that readers only interested in the homogeneous model can skip forward to Section 2.5 for a discussion of the eigenvalues and dimensions of the Jordan blocks.

The suggestion to present all calculations in the folded representation is a useful one, but when writing the manuscript we made the decision to keep all calculations as explicit as possible. While the folded representation would definitely simplify some expressions, we thought it might also obscure some calculations and make the connection with some of the established literature on integrability less clear, which is why we have chosen to present these results in this way.

---

## Editorial Decision

published